

# MeshXL: Neural Coordinate Field for Generative 3D Foundation Models

**Sijin Chen**[1,2,*]**, Xin Chen**[2,†]**, Anqi Pang**[2]**, Xianfang Zeng**[2]**, Wei Cheng**[2]**, Yijun Fu**[2]**,
Fukun Yin**[1,2]**, Zhibin Wang**[2]**, Jingyi Yu**[3]**, Gang Yu**[2]**, Bin Fu**[2]**, Tao Chen**[1,‡]

https://github.com/OpenMeshLab/MeshXL

[1]Fudan University    [2]Tencent PCG    [3]ShanghaiTech University

[†] project lead    [‡] corresponding author

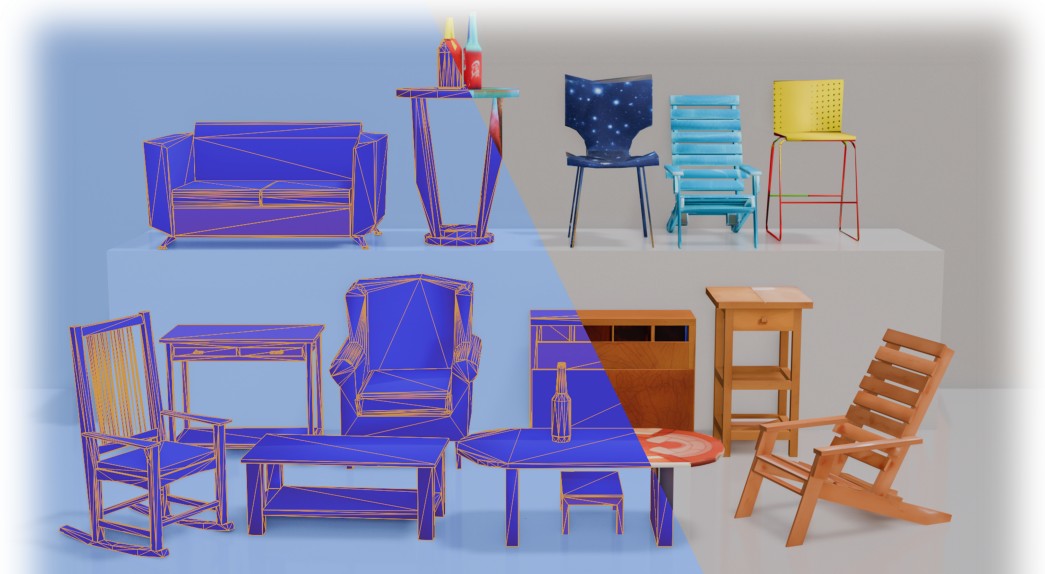

Figure 1: **MeshXL** can auto-regressively generate high-quality 3D meshes. We validate that **Neur**al **C**oordinate **F**ield (NeurCF), an explicit coordinate representation with implicit neural embeddings, is a simple-yet-effective sequence representation for large-scale mesh modelling.

## Abstract

The polygon mesh representation of 3D data exhibits great flexibility, fast rendering speed, and storage efficiency, which is widely preferred in various applications. However, given its unstructured graph representation, the direct generation of high-fidelity 3D meshes is challenging. Fortunately, with a pre-defined ordering strategy, 3D meshes can be represented as sequences, and the generation process can be seamlessly treated as an auto-regressive problem. In this paper, we validate **Neur**al **C**oordinate **F**ield (NeurCF), an explicit coordinate representation with implicit neural embeddings, is a simple-yet-effective representation for large-scale sequential mesh modeling. After that, we present MeshXL, a family of generative pre-trained auto-regressive models that addresses 3D mesh generation with modern large language model approaches. Extensive experiments show that MeshXL is able to generate high-quality 3D meshes, and can also serve as foundation models for various down-stream applications.

---

[*]Research done when Sijin Chen was a research intern at Tencent PCG.

38th Conference on Neural Information Processing Systems (NeurIPS 2024).

# 1  Introduction

The generation of high-quality 3D assets [61, 77, 29] is essential for various applications in video games, virtual reality, and robotics. Among existing 3D representations [51, 38, 57, 61], 3D mesh represents 3D data as graphs, which possesses the flexibility and accuracy representing sharp edges as well as both flat and curved surfaces. However, the direct generation of high-quality 3D meshes is challenging, given 1) the unstructured graph representation and 2) the demand for estimating accurate spatial locations and connectivity within vertices.

To generate 3D meshes, many works adopt an indirect way by first producing data in other 3D representations, such as point clouds [97, 49, 54], SDF [88, 94], and multi-view images [46, 82, 30]. After that, they adopt re-meshing methods [37] to post-process the generated geometries. There are also attempts towards the direct generation of 3D polynomial meshes. PolyGen [53] adopts two separate decoder-only transformers for vertices generation and vertices connectivity prediction. MeshGPT [65] first builds a mesh VQVAE to first turn meshes into tokens, and then learns to generate the token sequences with a GPT model [59]. Meanwhile, PolyDiff [2] directly adopts discrete denoising diffusion [4] on the discretized mesh coordinates.

Though these methods have achieved initial success in creating 3D assets, they suffer from certain limitations. To preserve sufficient high-frequency information, point clouds and voxels requires dense samplings on the object surfaces, which inevitably leads to great redundancy while representing flat surfaces. The reconstruction-based methods [82, 30, 67], however, rely heavily on the accuracy of the multi-vew generation pipelines [46]. Additionally, the VQVAE-based 3D generation methods [88, 65] require sophisticated multi-stage training, which less favors learning from large scale data.

To tackle the above challenges and explore the potential of scaling up 3D generative pre-training, we introduce a simple-yet-effective way of 3D mesh representation, the **Neur**al **C**oordinate **F**ield (NeurCF). NeurCF represents the explicit 3D coordinates with implicit neural embeddings. We show that with a pre-defined ordering strategy, a 3D mesh can be represented by a one-and-only coordinate sequence, which further helps us formulate 3D mesh generation as an auto-regressive problem. After that, we present MeshXL, a family of generative pre-trained transformers [93, 59], for the direct generation of high-fidelity 3D meshes. Without resorting to intermediate 3D representations, NeurCF facilitates an end-to-end learning pipeline for the direct pre-training on large-scale 3D mesh data.

By organizing high-quality 3D assets from ShapeNet [9], 3D-FUTURE [22], Objaverse [17], and Objaverse-XL [16], we achieve a collection of over 2.5 million 3D meshes to support large-scale generative pre-training. Extensive experiments demonstrate that the NeurCF representation facilitates MeshXL to generate higher-quality 3D meshes. By training on the collection of large-scale 3D mesh data, MeshXL can achieve better performance with larger numbers of parameters (Fig. 3 and Tab. 5), and surpass prior arts on multiple categories in the ShapeNet dataset [9] (Tab. 3).

In summary, our contributions can be summarized as follows:

- We validate that Neural Coordinate Field is a simple-and-effective representation of 3D mesh, which is also friendly to large-scale auto-regressive pre-training.

- We present a family of MeshXLs that can be treated as strong base models for image-conditioned or text-conditioned 3D mesh generation tasks.

- We show that MeshXL surpasses state-of-the-art 3D mesh generation methods, and can produce delicate 3D meshes compatible with existing texturing methods.

# 2  Related Work

First, we present a concise review of existing 3D representations. Subsequently, we discuss related works on 3D generation and recent efforts in developing 3D foundation models.

**3D Representations.** Researchers have long sought for accurate and efficient methods to represent 3D data. **Point Cloud** [54, 57, 58, 89] captures the spatial positions of discrete points in the Euclidean space, which is preferred by various 3D sensors [15, 87, 66, 3, 7]. **Mesh** [53, 2, 65, 12] represents the 3D structure with graphs. By connecting the vertices with edges, mesh can also be interpreted into a set of polygons in the 3D space. Similar to point clouds, **3D Gaussians** [38, 68] also record the discrete Euclidean distribution in 3D space. However, each point is represented by a 3D Gaussian

distribution function parameterized by its covariance matrix, color, and opacity. Given their fast convergence and rendering speed, 3D gaussians are often utilized for 3D reconstruction. **Neural Radiance Field** (NeRF) [51, 5] constructs a learnable volumetric function $f$ using neural networks trained on multi-view images. Due to its derivability and flexibility, NeRF is also favored for 3D generative models [46, 99, 76, 56]. Additionally, there are other 3D representations such as multi-view images [74, 90, 100], voxel fields [61, 13, 45], and signed distance fields [94], among others [64, 88, 63]. In this paper, we consider the **Neural Coordinate Field** (NeurCF), an explicit spatial representation with implicit neural embeddings, and investigate its potential for scalable 3D asset generation.

**3D Generation.** With the exploration of various 3D representations and the collection of large-scale 3D datasets [17, 9, 16], researchers have also put much effort exploring the generation of high-fidelity 3D assets [42, 39]. The **G**enerative **A**dversarial **N**etwork (GAN) [25, 80, 1, 33] produces synthetic 3D data with a generator $\mathcal{G}$, and train a discriminator network $\mathcal{D}$ to distinguish the generated and real data. Additionally, the potential of **diffusion** models [54, 28, 62] in the direct generation of 3D data is also widely explored [97, 2, 54, 50, 47]. The key idea behind diffusion is to transform the desired data distribution into a simpler distribution (*e.g.* gaussian) and learn a desnoising model for the reverse process. Besides, researchers have also explored the potential of diffusion models in generating **multi-view** images [46, 16, 82, 43], and reconstruct them into 3D structures. In this paper, we mainly explore the **auto-regressive** methods for 3D generation. AutoSDF [52] and MeshGPT [65] learn to generate discrete tokens and reconstruct them into 3D representations with a VQVAE model [72]. PolyGen [53] adopts two decoder-only transformers that predict the location and connectivity of vertices, sequentially. In this paper, we explore the potential of an explicit sequential modelling method for 3D meshes, and present a family of generative pre-trained transformers, MeshXL, for high-fidelity 3D mesh generation.

**3D Foundation Models.** The collection of large-scale high-quality 3D data [17, 16, 9, 81, 71, 21, 22] builds up the foundation for various 3D-related tasks [83, 27, 10, 41]. To explore the scaling effects in 3D learning, researchers have made great endeavors in building 3D foundation models for 3D understanding [96, 44, 98, 85, 86, 92, 100], reconstruction [30, 78, 67, 46, 16, 84, 73], and generation [61, 29, 65, 8]. With the introduction of large-scale 3D data in both variety and granularity [34, 41, 16], existing 3D foundation models are capable of generalizing to unseen concepts [100, 86, 44], generating high-fidelity 3D assets [88, 36, 65], responding to complex instructions [31, 10, 32, 41], and generating actions that interacts with the 3D environments [20, 79, 95]. In this paper, we present a fully end-to-end 3D mesh generation pipeline, explore the scaling effect for large-scale pre-training, and test whether our method can serve as a well-trained foundation model for various down-stream tasks.

# 3 Data

**Data Sources.** We provide details on the 3D data collections we use to train and evaluate our models. The whole data collection is built upon four widely-acknowledged 3D mesh datasets, *i.e.* ShapeNet V2 [9], 3D-FUTURE [22], Objaverse [17], and Objaverse-XL [16].

- **ShapeNet V2 [9]** collects about 51k 3D CAD models for 55 categories. We split the data in 9:1 for training and validation by each category.

- **3D-FUTURE [22]** present about 10k high-quality 3D mesh data for indoor furniture. However, because of the delicate design, the objects contain many faces. Therefore, only a small proportion of the data can be used to train our MeshXL models.

- **Objaverse [17]** is a large 3D data collection with more than 800k 3D objects for about 21k categories collected from Sketchfab. We split the data in 99:1 for training and validation, respectively.

- **Objaverse-XL [16]** further expand Objaverse [17] into a dataset with more than 10M 3D objects with additional data collected from GitHub, Polycam, Thingiverse, and Smithsonian. We split the Github and Thingiverse part of the Objaverse-XL dataset into 99:1 for training and validation.

**Data collection and filtering.** To organize existing datasets, we build up a filtering and pre-processing pipeline to ensure that the meshes meet our demand. We first collect meshes with fewer than 800 faces, and ensure that they have corresponding UV maps for rendering. After that, we render the 3D

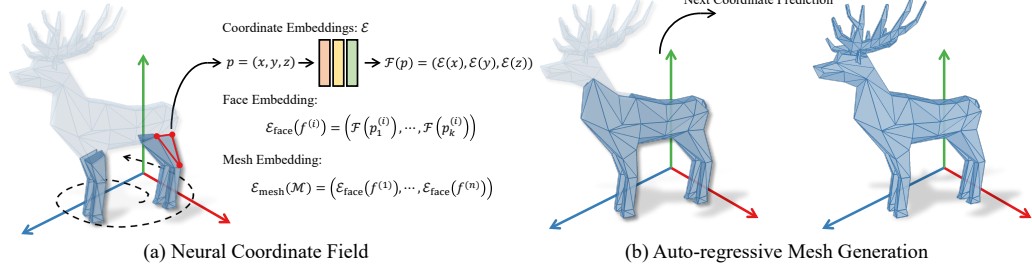

(a) Neural Coordinate Field  (b) Auto-regressive Mesh Generation

Figure 2: **Mesh Representation.** We present the **Neur**al **C**oordinate **F**ield (NeurCF) to encode the discretized coordinates in the Euclidean space. Benefiting from NeurCF and a pre-defined ordering strategy, our proposed MeshXL can directly generate the unstructured 3D mesh auto-regressively.

meshes, and discard those not center-aligned or occupying less than 10% of the rendered image. For those 3D objects with more than 800 but less than 20,000 faces, we use planar decimation to simplify the meshes. Finally, we achieve approximately 2.5 million 3D meshes (Tab. 1).

**Planar Decimation Pipeline.** To ensure the quality of the decimated 3D meshes, we make sure either a lower Hausdorff distance $\delta_{\text{hausdorff}}$ [65] or a similar rendered views [11].

**Collecting text-mesh pairs.** We render the 3D meshes with 12 different views and use CogVLM [75] to annotate 1) the front view and 2) the concatenated multi-view image. Then, we adopt the Mistral-7B-Instruct model [35] with in-context examples to generate a fused mesh caption.

**Data Statistics.** We present the data statistics of our large-scale 3D mesh collection in Tab. 1. After organizing and combing 3D assets from ShapeNet [9], 3D-FUTURE [22], Objaverse [17], and Objaverse-XL [16], we could achieve a total of 2.5 million 3D meshes.

Table 1: **Statistics for the Training Data and Validation Data.** After combining four data sources, our proposed MeshXL models are trained on approximately 2.5 million 3D meshes.

| Dataset | Pre-training | | Text-to-3D | |
|---|---|---|---|---|
| | Train | Val | Train | Val |
| ShapeNet [9] | 16,001 | 1,754 | 15,384 | 1,728 |
| 3D-Future [22] | 1,603 | - | - | - |
| Objaverse [17] | 85,282 | 854 | 83,501 | 820 |
| Objaverse-XL [16] | 2,407,337 | 15,200 | 1,347,802 | 13,579 |
| **Total** | 2,510,223 | 17,808 | 1,446,678 | 16,127 |

## 4 Neural Coordinate Field

**Neur**al **C**oordinate **F**ield (NeurCF) is an explicit representation with implicit neural embeddings. To be specific, for a Euclidean 3D coordinate system, we can partition the vertices coordinates into an $N^3$ grid. Then, each discretized coordinate $p = (x, y, z)$ can be encoded with the coordinate embedding layer $\mathcal{E}$, where $\mathcal{F}(p) = (\mathcal{E}(x), \mathcal{E}(y), \mathcal{E}(z))$. Therefore, a $k$-sided polynomial face $f^{(i)}$ can be encoded with $\mathcal{E}_{\text{face}}(f^{(i)}) = (\mathcal{F}(p_1^{(i)}), \cdots, \mathcal{F}(p_k^{(i)}))$. For simplicity, we share the learnable coordinate embeddings $\mathcal{E}$ among axes.

**Ordering.** Due to the graph representation, the order of the mesh vertices and the order of the edges among them are permutation-invariant. A pre-defined ordering strategy is essential to facilitate the sequence modelling in MeshXL. We employ the same ordering strategy as PolyGen [53] and MeshGPT [65]. The mesh coordinates are first normalized into a unit cube based on the mesh's longest axis, and discretized into unsigned integers. Within each face, the vertices are cyclically permuted based their coordinates ($z$-$y$-$x$ order, from lower to higher), which helps to preserve the direction of normal vectors. Then, we order these faces based on the permuted coordinates (lower to high). To this end, we can represent each 3D mesh with a **one-and-only** coordinate sequence, aiding large-scale generative pre-training on a large collection of 3D mesh data. With the NeurCF

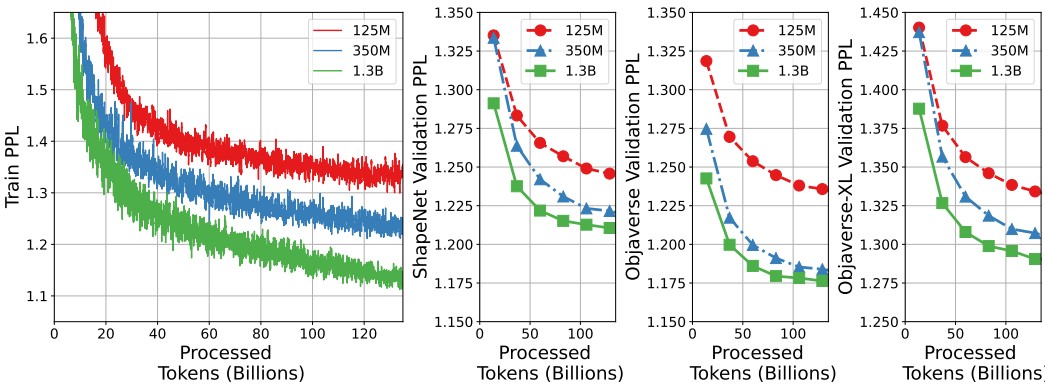

Figure 3: **Training and Validation Perplexity (PPL) for MeshXL Models.** We train all the models from scratch on 150 billion tokens. We observe that the performance grows with model sizes.

representation, an $n$-faced 3D $k$-sided polynomial mesh can be represented as the coordinate sequence $\mathcal{M} \in \mathbb{Z}^{n \times k \times 3}$, and further be encoded into $\mathcal{E}_{\text{mesh}} = (\mathcal{E}_{\text{face}}(f^{(1)}), \cdots, \mathcal{E}_{\text{face}}(f^{(n)}))$.

**A Sequential Mesh Representation.** One direct way to represent the 3D meshes is to directly reshape $\mathcal{M}$ into a vector with $(n \cdot k \cdot 3)$ tokens. As a special case, an $n$-faced triangular mesh can be represented by a vector with $9n$ tokens. Meanwhile, our representation can also be expanded to hybrid polynomial mesh representations with the proper introduction of separate tokens. For example, we can generate triangles within "<tri> $\cdots$ </tri>" and quadrilaterals within "<quad> $\cdots$ </quad>" also in one sequence. To identify the start and end of a mesh sequence, we add a <bos> ("begin-of-sequence") token before the mesh sequence and an <eos> ("end-of-sequence") token after.

**Comparisons.** Since we represent each coordinate with learnable embeddings, NeurCF is an end-to-end trainable representation for unstructured 3D meshes. Comparing to the decoupled vertex and polygon representation in PolyGen [53], NeurCF only requires one coordinate sequence for each 3D mesh. Additionally, NeurCF is storage and computation efficient comparing to voxel fields ($O(N^3)$), since it can easily scale up the resolution with a complexity of $O(N)$.

## 5 Method

We first present the architecture and training objective for MeshXL models. Then, we show that MeshXL models can take an additional modality as the condition for controllable 3D assets generation. After this, we investigate the effects of scaling.

**Architecture.** In Sec. 4, we present a simple-yet-effective way to represent a 3D mesh into a sequence. Therefore, the learning of 3D mesh generation can be formulated as an auto-regressive problem, which can be seaminglessly addressed by modern **L**arge **L**anguage **M**odel (LLM) approaches. In our paper, we adopt the decoder-only transformers using the OPT [93] codebase as our base models. To adapt the pre-trained OPT models to our *next-coordinate prediction* setting, we fine-tune the whole model with newly-initialized coordinate and position embeddings.

**Generative Pre-Training.** We train MeshXL models using the standard next-token prediction loss. Given the trainable weights $\theta$ and an $|s|$-length sequence $s$, the generation loss is calculated as:

$$\mathcal{L}_{\text{MeshXL}}(\theta) = -\sum_{i=1}^{|s|} \log P\left(s_{[i]} | s_{[1, \cdots, i-1]}; \theta\right). \quad (1)$$

For each mesh sequence, we add a <bos> token before the mesh tokens, and an <eos> token after to identify the ending of a 3D mesh. During inference, we adopt the top-$k$ and top-$p$ sampling strategy to produce diverse outputs.

$\mathcal{X}$**-to-Mesh Generation.** Here we mainly consider generating 3D meshes from images and texts. We first turn the extra conditions into tokens with pre-trained encoders [18, 19]. To align the additional

text/image feature with the mesh coordinate field, we adopt the Q-Former architecture [40] to compress the encoded feature into a fixed-length of 32 learnable tokens as the prefix of the MeshXL model. The overall training objective for the conditional mesh generation is shown in Eq. (2):

$$\mathcal{L}_{\mathcal{X}\text{-to-mesh}}(\theta) = -\sum_{i=1}^{|s|} \log P\left(s_{[i]}|s_{[1,\cdots,i-1]};\mathcal{X}\right).\tag{2}$$

During inference, the model predicts the mesh tokens after the fixed-length prefix.

**Scaling Up.** We present MeshXL in various sizes, including 125M, 350M, and 1.3B. The detailed hyperparameters for training different models can be found in Tab. 2. To better analyze the scaling effects, we train all models from scratch on 150 billion tokens. We provide both training curve and validation perplexity for different models in Fig. 3. One can see that as the number of parameters grows, the model achieves a lower validation perplexity, indicating a higher probability to produce the validation data.

Table 2: **Hyperparameters for different MeshXL Base Models.** We present three MeshXL models with 125M, 350M, and 1.3B parameters, respectively.

| Hyperparameters | MeshXL(125M) | MeshXL(350M) | MeshXL(1.3B) |
|---|---|---|---|
| # Layers | 12 | 24 | 24 |
| # Heads | 12 | 16 | 32 |
| $d_{\text{model}}$ | 768 | 1,024 | 2,048 |
| $d_{\text{FFN}}$ | 3,072 | 4,096 | 8,192 |
| Optimizer | AdamW($\beta_1$=0.9, $\beta_2$=0.999) | | |
| Learning rate | $1.0 \times 10^{-4}$ | $1.0 \times 10^{-4}$ | $1.0 \times 10^{-4}$ |
| LR scheduler | Cosine | Cosine | Cosine |
| Weight decay | 0.1 | 0.1 | 0.1 |
| Gradient Clip | 1.0 | 1.0 | 1.0 |
| Number of GPUs | 8 | 16 | 32 |
| # GPU hrs (A100) | 1,944 | 6,000 | 23,232 |

# 6 Experiments

We first briefly introduce the data, metrics, and implementation details in Sec. 6.1. Then, we provide evaluations and comparisons on the generated meshes ($cf$. Sec. 6.2) and ablations ($cf$. Sec. 6.3). We also provide visualization results in Sec. 6.4.

## 6.1 Data, Metrics, and Implementation Details

**Data.** We pre-train the base model with 2.5 million 3D meshes collected from the combination of ShapeNet [9], 3D-FUTURE [22], Objaverse [17], and Objaverse-XL [16]. We use planar decimation on meshes with more than 800 faces following MeshGPT [65] and RobustLowPoly [11]. For generative mesh pre-training, we randomly rotate these meshes with degrees from ($0°$, $90°$, $180°$, $270°$), and adopt random scaling along each axis within range $[0.9, 1.1]$ for data augmentation.

**Metrics.** We follow the standard evaluation protocols in MeshGPT [65] and PolyDiff [2] to measure the quality of the generated meshes with the following metrics. We use Coverage (COV) to quantify the diversity of the generated meshes, which is sensitive to mode dropping but cannot be used to assess the generation quality. Minimum Matching Distance (MMD) calculates the average distance between the reference set and their closest neighbors in the generated set, but is not sensitive to low-quality results. The 1-Nearest Neighbor Accuracy (1-NNA) quantifies the quality and diversity between the generation set and the reference set, whose optimal value is 50%. We also adopt the Jensen-Shannon Divergence (JSD) score. Among all the above metrics, we use Chamfer Distance to measure the similarity between two samples. We also render the generated meshes and adopt the Frechet Inception Distance (FID) and Kernel Inception Distance (KID) on the rendered images for feature-level evaluation. We multiply the MMD, JSD, and KID scores by $10^3$.

**Implementation.** We conduct all the experiments on a cluster consisting 128 A100 GPUs. We train our models under bfloat16 with the ZeRO-2 strategy [60] using the AdamW [48] optimizer with a

Table 3: **Quantitative Comparisons with Prior Arts on ShapeNet [9].** We scale MMD, JSD, KID by $10^3$. MeshXL can produce diverse and high-quality 3D meshes.

| Category | Methods | COV↑ | MMD↓ | 1-NNA | JSD↓ | FID↓ | KID↓ |
|----------|---------|------|------|-------|------|------|------|
| Chair | PolyGen [53] | 7.79 | 16.00 | 99.16 | 228.80 | 63.49 | 43.73 |
| | GET3D [23] | 11.70 | 15.92 | 99.75 | 155.25 | 67.84 | 42.10 |
| | MeshGPT [65] | 42.00 | 4.75 | 69.50 | 55.16 | 39.52 | 8.97 |
| | MeshXL (125M) | 50.80 | **3.11** | 56.55 | 9.69 | 28.15 | 1.48 |
| | MeshXL (350M) | 50.80 | 3.17 | **55.80** | 9.66 | 28.29 | **1.39** |
| | MeshXL (1.3B) | **51.60** | 3.23 | **55.80** | **9.48** | **9.12** | 1.84 |
| Table | PolyGen [53] | 44.00 | 3.36 | 67.20 | 25.06 | 54.08 | 14.96 |
| | GET3D [23] | 16.80 | 10.39 | 91.90 | 226.97 | 67.65 | 34.62 |
| | MeshGPT [65] | 34.30 | 6.51 | 75.05 | 92.88 | 53.75 | 7.75 |
| | MeshXL (125M) | 51.21 | 2.96 | 57.96 | **12.82** | 42.55 | **0.92** |
| | MeshXL (350M) | 49.70 | 3.07 | **56.10** | 13.64 | 43.43 | 1.27 |
| | MeshXL (1.3B) | **52.12** | **2.92** | 56.80 | 14.93 | **22.29** | 2.03 |
| Bench | PolyGen [53] | 31.15 | 4.01 | 83.23 | 55.25 | 70.53 | 12.1 |
| | MeshGPT [65] | 34.92 | 2.22 | 68.65 | 57.32 | 52.47 | 6.49 |
| | MeshXL (125M) | 54.37 | 1.65 | **43.75** | 16.43 | **35.31** | **0.82** |
| | MeshXL (350M) | 53.37 | 1.65 | 42.96 | **15.41** | 36.35 | 0.96 |
| | MeshXL (1.3B) | **56.55** | **1.62** | 39.78 | 15.51 | 35.50 | 1.60 |
| Lamp | PolyGen [53] | 35.04 | 7.87 | 75.49 | 96.57 | 65.15 | 12.78 |
| | MeshGPT [65] | 41.59 | 4.92 | 61.59 | 61.82 | 47.19 | 5.19 |
| | MeshXL (125M) | **55.86** | 5.06 | 48.24 | 43.41 | 34.61 | **0.84** |
| | MeshXL (350M) | 53.52 | **4.18** | 49.41 | **34.87** | **25.94** | 1.92 |
| | MeshXL (1.3B) | 51.95 | 4.89 | 47.27 | 41.89 | 31.66 | 0.99 |

learning rate decaying from $10^{-4}$ to $10^{-6}$ and a weight decay of 0.1. The detailed hyperparameters for different models can be found in Tab. 2. To train our base models, we load the weights from the pre-trained OPT models [93] and initialize the word embeddings and positional embeddings from scratch. Without further specification, we generate 3D meshes with the top-$k$ and top-$p$ sampling strategy with $k = 50$ and $p = 0.95$.

## 6.2 Evaluations and Comparisons

We provide quantitative as well as qualitative comparisons on both unconditional and conditional 3D mesh generation on public benchmarks.

**Unconditional Generation.** We evaluate MeshXL as well as other baseline methods using the ShapeNet [9] data in Tab. 3. We split the data by 9:1 for training and validation by each category. For evaluation, we fine-tune our pre-trained base model and sample 1,000 meshes for each category. Among the listed methods, we reproduce the MeshGPT [65] with a GPT2-medium model (355M) [59]. With a similar number of parameters, Mesh-XL (350M) out-performs MeshGPT by a large margin, showing a higher COV score, a lower MMD score, and a closer 1-NNA score to 50%. This indicates that MeshXL can produce diverse and high-quality 3D meshes.

Table 4: **User Study.** Compared to baseline methods, the meshes generated by MeshXL are better aligned with human preference in terms of both geometry and designs.

| Methods | Quality↑ | Artistic↑ | Triangulation↑ |
|---------|----------|-----------|----------------|
| PolyGen [53] | 2.53 | 2.72 | 3.15 |
| GET3D [23] | 3.15 | 2.46 | 3.15 |
| MeshXL | **3.96** | **3.45** | **3.72** |
| Reals | 4.08 | 3.33 | 3.75 |

**User Study.** To evaluate how well the generated 3D meshes align with human preference, we perform user studies on the chair category in Tab. 4 with several baseline methods [53, 23]. We recruit and instruct the participants to score each mesh from 0 to 5 (higher is better) based on its 1) **quality**: the smoothness of object surfaces and completeness of the mesh, 2) **artistic**: how much do you believe this object is designed and created by artists, and 3) **triangulation**: how well do the connectivity among vertices aligns with the models created by professional designing software [14]. As a baseline evaluation, we also ask the participants to score the ground truth 3D geometries sampled from the

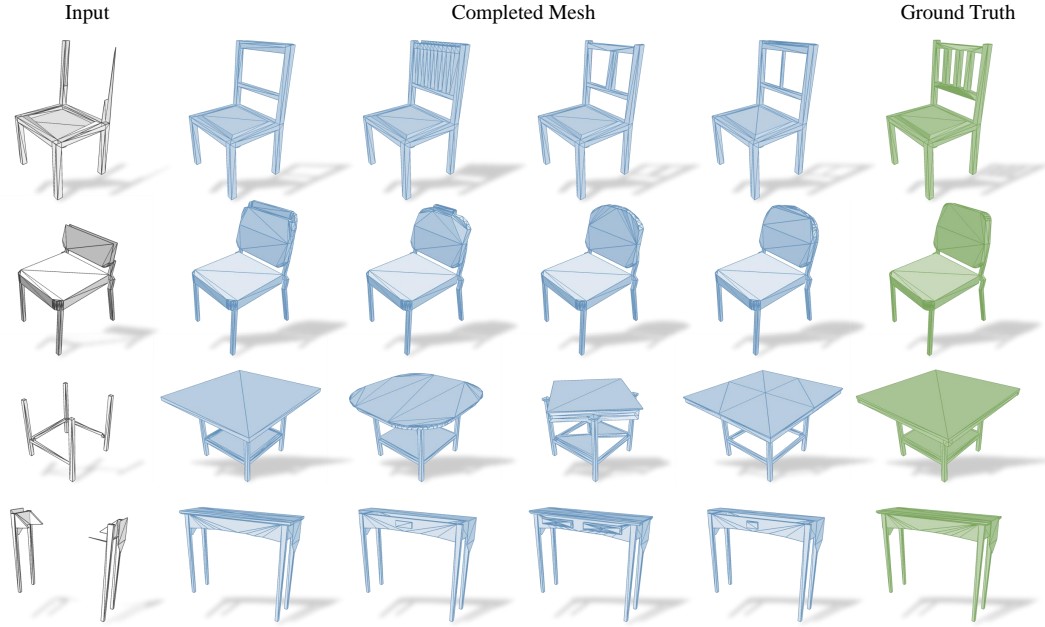

| Input | Completed Mesh | Ground Truth |

Figure 4: **Evaluation of Partial Mesh Completion.** Given some partial observation of the 3D mesh (white), MeshXL is able to produce diverse object completion results (blue).

ShapeNet data. We have collected a total of 434 valid responses. The results show that the 3D meshes created by MeshXL are consistently preferred by human in all dimensions.

## 6.3  Ablation Studies

**Necessity of Mesh VQVAE.** Comparing to MeshGPT [65], MeshXL is an end-to-end trainable model that produces 3D meshes with *next-coordinate prediction*. We show in Tab. 3 that, MeshXL out-performs MeshGPT with similar numbers of parameters. We also show that MeshXL can produce high quality 3D meshes with both sharp edges and smooth surfaces in Fig. 7. Furthermore, MeshXL can save the effort training a vector quantized mesh tokenizer [65, 72], which further facilitates generative pre-training on large scale datasets.

**Effectiveness of Model Sizes.** To analyze whether pre-training a larger model benefits 3D mesh generation, we evaluate MeshXL base models with different sizes on the Objaverse [17] dataset in Tab. 5. We observe that as the model size grows, the generated samples exhibits a larger COV, smaller JSD score, and a closer 1-NNA to 50%, which indicates an improved diversity and quality.

Table 5: **Effectiveness of Model Sizes on Objaverse.** As the model size grows, MeshXL achieves a closer 1-NNA to 50%, a larger COV and a smaller JSD, indicating better diversity and quality.

| Method | COV↑ | MMD↓ | 1-NNA | JSD↓ | FID↓ | KID↓ |
|---|---|---|---|---|---|---|
| MeshXL (125M) | 39.76 | 5.21 | 67.34 | 26.03 | 17.32 | 4.48 |
| MeshXL (350M) | 40.79 | 5.20 | 65.68 | 23.71 | 15.14 | 3.33 |
| MeshXL (1.3B) | **42.86** | **4.16** | **61.56** | **20.99** | **12.49** | **2.94** |

**Shape Completion.** To analysis whether our method is capable of producing diverse outputs, we adopt MeshXL (1.3B) model to predict the whole object given some partial observations. In practice, we use 50% of the object mesh as input, and ask the model to predict the rest 50% in Fig. 4. One can see that Mesh-XL is able to produce diverse and reasonable outputs given the partial observation of the 3D mesh.

$\mathcal{X}$**-to-Mesh Generation.** To adopt the MeshXL base models to the $\mathcal{X}$-to-mesh generation setting, we adopt the Q-Former [40] to encode the additional conditions as prefixes. We showcases several conditional generation results in Fig. 5. We show that MeshXL can generate high-quality 3D meshes given the corresponding image or text as the additional inputs.

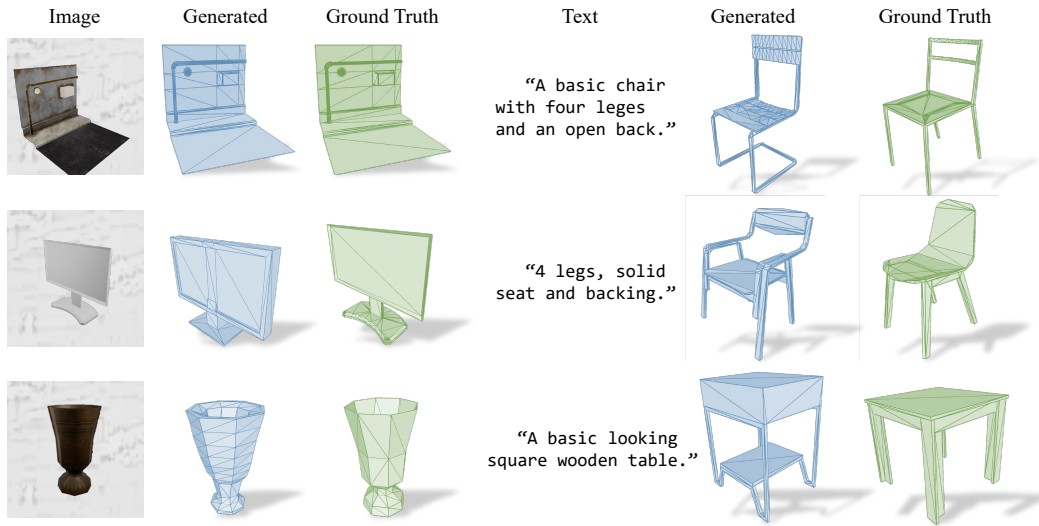

Figure 5: **Evaluation of $\mathcal{X}$-to-mesh generation.** We show that MeshXL can generate high-quality 3D meshes given the corresponding image or text as the additional inputs.

**Texturing.** We adopt Paint3D [91], a coarse-to-fine texture generation pipeline, to generate textures for the 3D meshes produced by MeshXL in Fig. 6. We show that 3D meshes produced by MeshXL can easily fit in existing texturing methods to produce high-quality 3D assets.

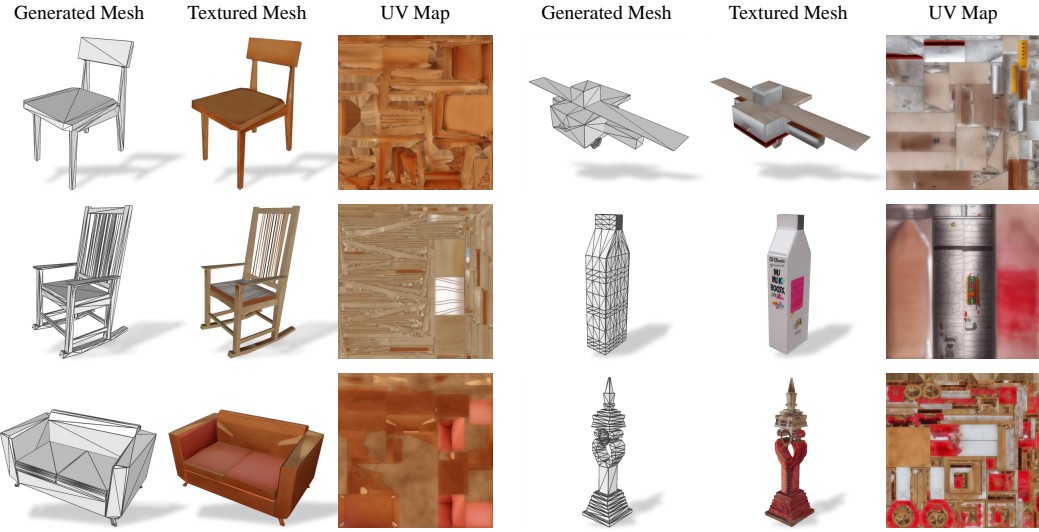

Figure 6: **Texture Generation for the Generated 3D Meshes.** We adopt Paint3D [91] to generate textures for 3D meshes produced by MeshXL.

## 6.4 Visualizations

We provide qualitative comparisons on the generated meshes in Fig. 7. MeshXL is able to produce high-quality 3D meshes with both sharp edges and smooth surfaces. We also visualize the normal vectors to compare the smoothness of object surfaces. The results show that 3D meshes generated by GET3D [23] have rough surfaces with tens of thousands of triangles, while MeshXL depicts the 3D shape with much smoother surfaces and less triangles.

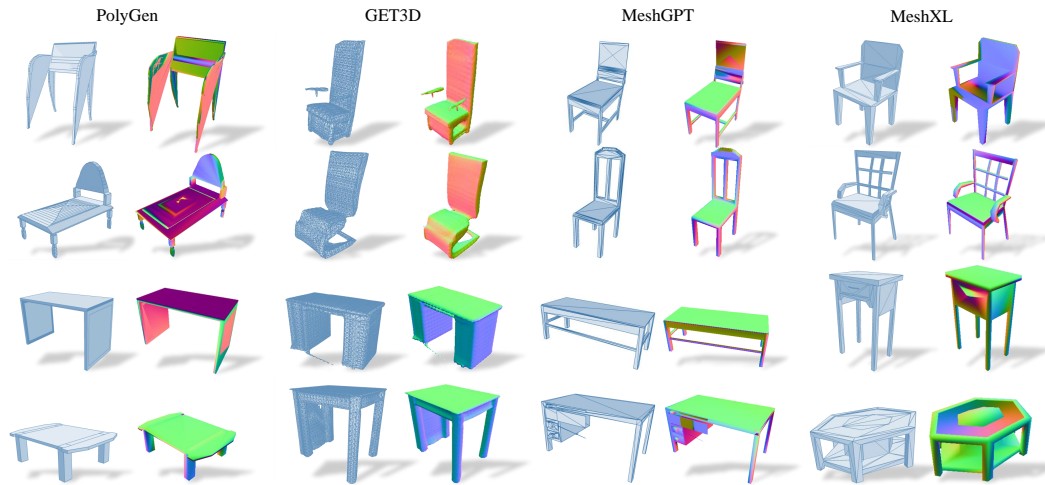

Figure 7: **Qualitative comparisons.** We visualize the generated meshes and normal vectors. MeshXL is able to produce high-quality 3D meshes with both sharp edges and smooth surfaces.

## 7 Discussions

**Difference with PolyGen** [53]. PolyGen treats 3D mesh data as a vertex sequence and a face sequence. PolyGen first generates the ordered vertices with the vertex transformer, then predicts the connectivity among vertices with the face transformer. Comparing to PolyGen, our proposed MeshXL adopts a more straightforward approach that *models the 3D mesh as a one-and-only coordinate sequence*, which further supports the direct and end-to-end pre-training on a large collection of 3D data.

**Difference with MeshGPT** [65]. MeshGPT consists of a mesh VQVAE [72] and a decoder-only transformer [59]. MeshGPT first learns a mesh VQVAE to quantize the 3D meshes into discrete tokens. After that, MeshGPT trains a decoder-only transformer to generate the discrete tokens for 3D mesh reconstruction. In comparison, our proposed MeshXL is an end-to-end method that learns the neural representation of coordinates and outputs 3D meshes directly.

**Extensibility.** Our method, MeshXL, is built upon the concept of auto-regressive methods. Therefore, our method is not restricted to the decoder-only transformers [59, 93, 69, 70], and can also be extended to other causal language models (*i.e.* Mamba [26], RWKV [55], and xLSTM [6]).

## 8 Limitations, Future Work, and Conclusions

**Limitations and Future Work.** The main drawback of MeshXLs is the inference time. During sampling, MeshXL will generate 7,200 tokens for an 800-faced 3D mesh, which takes a relatively long time because of the auto-regressive process. As for future works, recent endeavors on the RNN-related methods [6, 55, 26] and multiple tokens prediction for LLMs [24] might open up great opportunities in saving the inference cost.

**Conclusion.** We validate that NeurCF, an explicit coordinate representation with implicit neural embeddings, is a simple-and-effective representation of 3D meshes. By modelling the 3D mesh generation as an auto-regressive problem, we seek help from modern LLM approaches and present a family of generative pre-trained models, MeshXL, for high-fidelity 3D mesh generation. We show that MeshXL performs better given larger-scale training data and increased parameters. Extensive results show our proposed MeshXL can not only generate high-quality 3D meshes, but also exhibits great potential serving as base models for conditional 3D assets generation.

## Acknowledgement

This work is supported by National Natural Science Foundation of China (No. 62071127), National Key Research and Development Program of China (No. 2022ZD0160101), Shanghai Natural Science Foundation (No. 23ZR1402900), Shanghai Municipal Science and Technology Major Project (No.2021SHZDZX0103).

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

# A  Appendix

## A.1  More Visualization Results

**Unconditional Results on ShapeNet.** We visualize unconditional 3D mesh generation results for chair, table, lamp and bench in Fig. 8. One can see that MeshXL is able to produce diverse and high-quality 3D meshes.

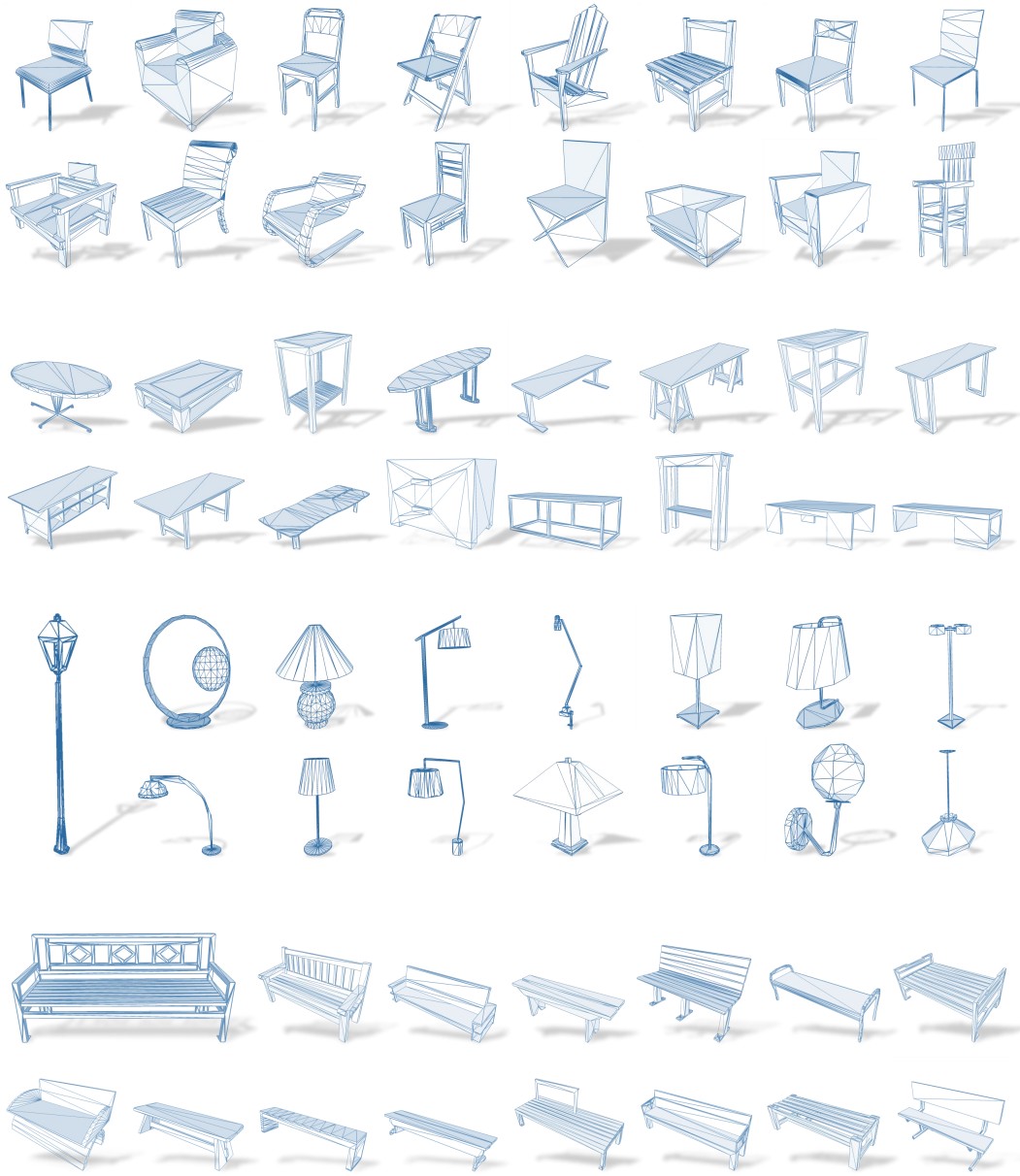

Figure 8: **Gallery results.** Additional generation results for chair, table, lamp, and bench.

**Unconditional Generation on Objaverse.** We visualize 3D meshes randomly sampled from MeshXL base model in Fig. 9. After training on a large-scale collection of 3D mesh data, MeshXL is able to produce diverse and high-quality 3D meshes.

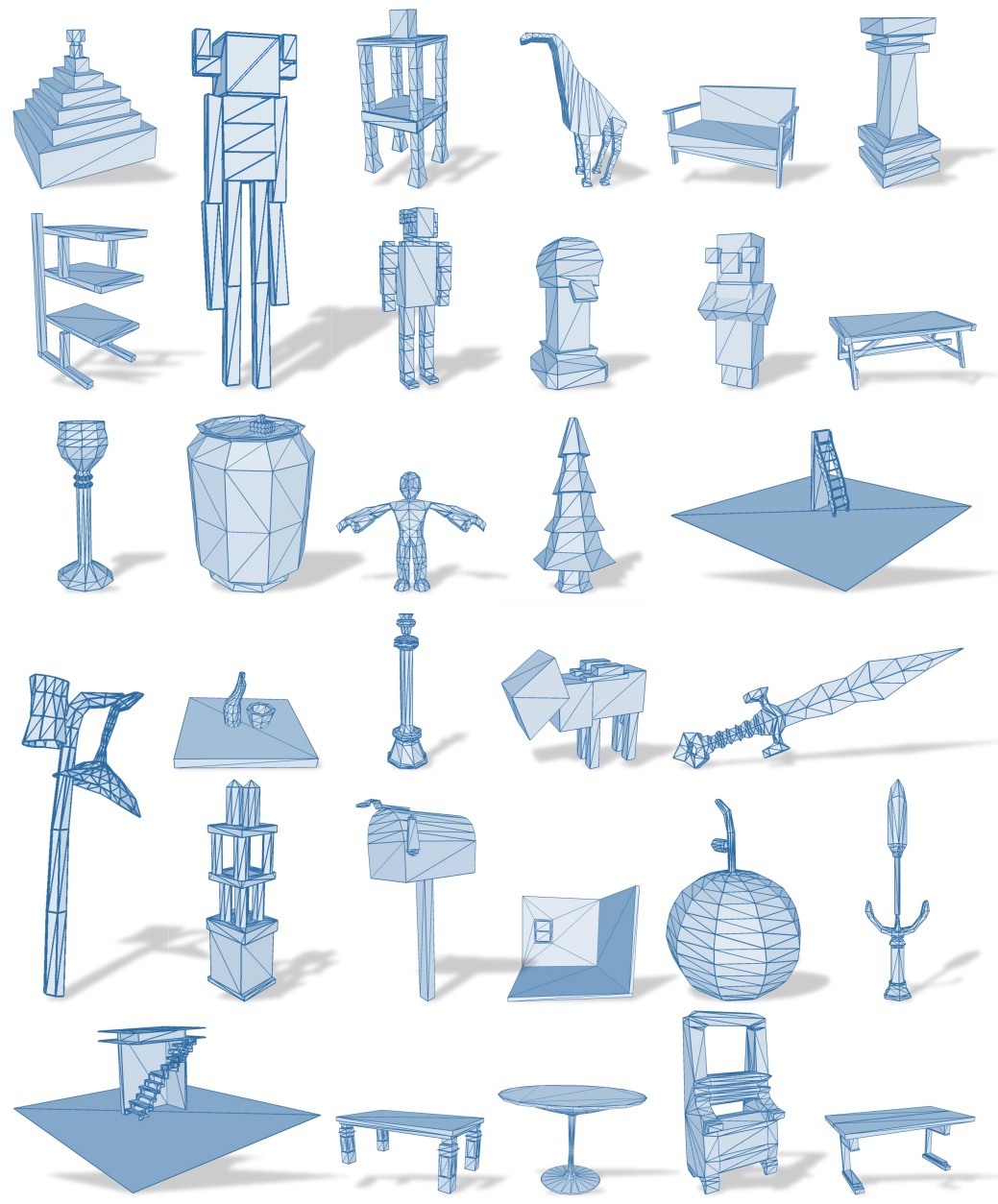

Figure 9: **Gallery results.** MeshXL is able to produce diverse 3D meshes with high quality.

## A.2 Mesh Quality Assessment

**How well is the triangulation.** We evaluate the aspect ratio, face area, and number of faces for a better evaluation in Tab. 6. Though the meshes generated by our MeshXL have a higher average aspect ratio, we manage to achieve a smaller variance with much less 3D faces. This indicates the stability of our generation ability and the efficiency of the direct mesh representation. Since we train our MeshXLs only on triangular meshes, long-thin triangles inevitably exist in our training data. By co-training our MeshXLs on triangular meshes, 3D quads, and even hybrid representations, we could reduce the existence of long thin triangles for better generation quality.

Table 6: **Mesh Quality Assessment.** We evaluate the aspect ratio, face area and number of faces for the generated 3D meshes.

| Method | Aspect Ratio | | Face Area | | Number of Faces | |
|---|---|---|---|---|---|---|
| | mean | std. | mean | std. | mean | std. |
| GET3D [23] | 6.27 | 116.03 | 0.000 | 0.000 | 27251.80 | 11535.135 |
| MeshXL (125M) | 10.47 | 16.88 | 0.031 | 0.096 | 327.34 | 174.53 |
| MeshXL (350M) | 10.25 | 16.09 | 0.032 | 0.099 | 342.24 | 193.97 |
| MeshXL (1.3B) | 10.23 | 15.91 | 0.034 | 0.102 | 320.36 | 195.43 |

## A.3 Inference Time Analysis

The inference cost of MeshXL is closely related to the numbers of generated faces and the model sizes. We perform inference cost analysis with a batch size of one using bfloat16 on a single RTX 3090. We carry out a an analysis (Tab. 7) on 1) the average inference time of generating a given number of triangles, and 2) the average inference time of generating 3D meshes.

Table 7: **Inference cost of MeshXL models.** We carry out inference cost analysis on time duration and memory usage under bfloat16 with a single RTX 3090.

| num faces | MeshXL (125M) | | MeshXL (350M) | | MeshXL (1.3B) | |
|---|---|---|---|---|---|---|
| | time (s) | GPU mem. (GB) | time (s) | GPU mem. (GB) | time (s) | GPU mem. (GB) |
| 100 | 6.30 | 1.59 | 11.30 | 2..98 | 12.08 | 8.41 |
| 200 | 12.50 | 1.65 | 22.70 | 3.20 | 24.03 | 9.17 |
| 400 | 25.21 | 1.85 | 45.81 | 3.78 | 48.09 | 11.17 |
| 800 | 49.88 | 2.28 | 92.19 | 5.74 | 96.49 | 21.66 |
| **avg.** | 29.49 | - | 44.65 | - | 49.43 | - |

