# OpenReview forum: "MeshXL: Neural Coordinate Field for Generative 3D Foundation Models"
_NeurIPS.cc/2024/Conference — NeurIPS 2024 poster_

### Official Review · Reviewer_AHoX · 2024-07-12

**Soundness:** 3
**Presentation:** 3
**Contribution:** 3
**Rating:** 6
**Confidence:** 4

**Summary:**

The paper is about 3D mesh generative models. The generative method is done by generating triangles one-by-one (auto-regressively). Each face is represented by 3 vertices (9 coordinates).

**Strengths:**

The idea to generate faces sounds interesting than previous methods. Unlike PolyGen which needs to generate connections (faces) between vertices, the method seems to be more elegant and easier for programming. The ShapeNet results are also convincing.

**Weaknesses:**

The major limitation of the method is the sequence length. The method is only able to handle 800 faces with the current resources. This makes the method is nearly impossible to generate complicated meshes.

The method also looks a bit brutal. Some faces share vertices. But the method needs to generate the shared vertices multiple times. The method can also generate intersecting faces.

**Questions:**

How is resolution defined? The number N (in L103) seems to be missing in the context. In L111, the authors mentioned it should be an unsigned integer. But it is still not clear what the number should be. In my experience, the number should be large (or the classification will be very difficult when training the autoregressive transformer). If the number is too small, then the resolution is limited.

I didn't find the sampling time and hardware. Sampling 7200 tokens takes lots of time. I would like to see a time analysis (and maybe memory consumption).

The training is done on Objaverse-XL but the main results are mainly about ShapeNet. The comparison is not fair. Also I can only find Fig 6 is about general objects instead of Shapenet objects. The quality seems to be very limited.

Even the authors call the method neural coordinate fields, but it is nothing about field. A field should be like a function with continuous domain (like nerf and signed distance fields). The method is similar to PolyGen. All vertices are discretized.

**Limitations:**

Yes

---

> ### Author Rebuttal · Authors · 2024-08-06
>
> 📝 **W1: The limitation of sequence length.**
>
> 💡**A:** The main focus of our paper is to establish end-to-end methods with less inductive bias and pave the way for scaling up learning from large-scale 3D data. We restrict the number of faces to 800 for better alignment with PolyGen and MeshGPT. However, by equipping MeshXL with modern LLM techniques, such as RoPE and ring attention, we are ready to extend our method to more complicate 3D meshes. Additionally, we will also be able to train larger MeshXL networks.
>
>
> 📝 **W2: Potential redundancy in the mesh representation and intersecting faces.**
>
> 💡**A:** **The redundancy in mesh representation paves the way for large-scale pre-training.** We acknowledge that the NeurCF representation does generate shared vertices multiple times. However, this enables us to represent each 3D mesh with **only one coordinate sequence** and design an **end-to-end** pipeline on large scale 3D data.
>
> **Analysis of potential surface artifacts.** The potential occurrence of surface artifacts could be the limitation for our MeshXL, and all existing auto-regressive mesh generation methods (PolyGen and MeshGPT) for now. However, after training directly on large-scale 3D meshes, MeshXL can already generate high-quality 3D mesh data under a high success rate. It is also possible to further improve MeshXL by treating traditional methods as reward models or filters to eliminate the surface artifacts.
>
>
> 📝 **Q1: The resolution of MeshXL.**
>
> 💡**A:** To align with MeshGPT, we choose the resolution of $N=128$ in all our experiments. However, our MeshXL adopts an architecture close to large language models, whereas we can easily extend MeshXL to a larger $N$ with a time and space complexity of $O(N)$ by simply learning a $N$-sized coordinate embeddings and an output classifier with $N$ output channels. Comparing to the $O(N^3)$ complexity in traditional hard coding algorithms, it is easier for us to train models with higher resolution. We will also release the models with larger resolution in the future.
>
>
> 📝 **Q2: Inference time analysis**
>
> 💡**A:** The inference time of our MeshXL is closely related to the **numbers of generated faces** and the **model sizes**. We perform inference time analysis with **a batch size of one using BFloat16 on a single RTX 3090**. We carry out a an analysis on 1) the average inference time of generating a given number of triangles.
> | | MeshXL - 125m | | | MeshXL - 350m | | | MeshXL - 1.3b | |
> |-|-|-|-|-|-|-|-|-|
> | #faces | time (s) | GPU Mem (GB) |   | time (s) | GPU Mem (GB) |   | time (s) | GPU Mem (GB) |
> | 100 | 6.30   | 1.59 | | 11.30 | 2.98 | | 12.08 | 8.41   |
> | 200 | 12.50 | 1.65 | | 22.70 | 3.20 | | 24.03 | 9.17   |
> | 400 | 25.21 | 1.85 | | 45.81 | 3.78 | | 48.09 | 11.17 |
> | 800 | 49.88 | 2.28 | | 92.19 | 5.74 | | 96.49 | 21.66 |
>
> We also show 2) the average inference time of generating 3D meshes.
> | | MeshXL - 125m | MeshXL - 350m | MeshXL - 1.3b |
> |-|-|-|-|
> |avg. time (s) | 29.49 | 44.65 | 49.43|
>
> 📝 **Q3: Fair comparisons and evaluations on larger datasets.**
>
> 💡 **A:** We reproduce MeshGPT with `gpt2-medium` (355m) and re-implement MeshXL (marked as MeshXL$^{ShapeNet}$) by first training the model on all ShapeNet categories before fine-tuning it to a specified category. 📊 Please refer to **the global rebuttal** for exact results. With a similar amount of parameters (350m), our method could consistently achieve better generation results with a higher COV score, a lower MMD score, and a closer 1-NNA score to 50%.
>
> Additionally, to study the effectiveness of large-scale pre-training, we conduct evaluations on Objaverse. We show that as the model size grows, MeshXL exhibits better 3D mesh generation quality, with a higher COV score, a lower MMD score, and a closer 1-NNA score to 50%.
> | Model | COV $\uparrow$ | MMD $\downarrow$ | 1-NNA | JSD $\downarrow$ | FID $\downarrow$ | KID $\downarrow$ |
> |-|-|-|-|-|-|-|
> | MeshXL - 125m | 39.76 | 5.21 | 67.34 | 26.03 | 17.32 | 4.48 |
> | MeshXL - 350m | 40.79 | 5.20 | 65.68 | 23.71 | 15.14 | 3.33 |
> | MeshXL - 1.3b   | **42.86** | **4.16** | **61.56** | **20.99** | **12.49** | **2.94** |
>
> Additionally, we have provided additional generation results in the **uploaded PDF file**. We have also open-sourced our code and pre-trained weights to the community.
>
>
>
> 📝 **Q4: Justification of Neural Coordinate Field.**
>
> 💡**A:** We thank the reviewer for pointing out this constructive suggestion.
> 1. **Justification of Neural Coordinate Field**. Neural coordinate is a representation that treats vertex coordinates as coordinate embeddings. Meanwhile, field should be a function within continuous domain. As we currently learn coordinate embeddings that uniformly spread along each axis, we will try learning interpolation functions (e.g. linear interpolation, gaussian interpolation, or B-spine basis functions) to extend our embeddings to the continuous domain. We will also try extending our method to the continuous domain by replacing the coordinate embeddings with sinusoid function. We will perform detailed experiments in our revision.
> 2. **Relation to the ordering in PolyGen**. We adopt the same vertex representation as PolyGen, but a **different mesh representation**. Our MeshXL and PolyGen both represent vertices with discrete coordinates. However, **MeshXL represents a 3D mesh only with an ordered sequence of coordinate**, while PolyGen decouples the vertices generation and polygon generation, and adopts two sequences for each mesh, i.e., the vertex sequence and the face index sequence to connect the generated vertices. Therefore, the potential redundancy in our representation in turn supports end-to-end training and better suits learning from large-scale 3D data.

---

### Official Review · Reviewer_CF59 · 2024-07-13

**Soundness:** 3
**Presentation:** 2
**Contribution:** 2
**Rating:** 5
**Confidence:** 5

**Summary:**

The paper proposes MeshXL, a mesh generation model based on the Neural Coordinate Fields(NeurCF), which encodes discretized coordinates of mesh vertices into a sequence of tokens.
Then a decoder only transformer is trained to generate meshes unconditionally/conditioned on modality.
The model is trained on multiple datasets for better performance.
In terms of quantitative and qualitative results, the method outperforms other baselines.

**Strengths:**

The method proposes an end-to-end transformer model for mesh generation based on its neural coordinate field.
On the performance side, it produces 3D meshes with better quality.

**Weaknesses:**

The weaknesses of the paper mainly lies in the technical part and the comparison part.
To my knowledge, the main difference between previous discrete mesh generation methods like MeshGPT is that MeshGPT first encodes the mesh with VQVAE and then trains a generation model, while the paper does it in a single stage.
The paper does produce great results but seems miss a reason why single stage can give performance.
In the introduction part (lines 21–28), the paper discusses previous generative methods but does not explicitly point out why they are missing.
I think adding more analysis can help readers understand the paper better.

I am not sure the comparisons between MeshXL and PolyGen/GET3D are fair in Table. 2 since it uses pre-trained MeshXL and fine-tunes it on a specific category. I believe using the same dataset for training is more convincing.

For reference, I think template-based deformation methods and command-based methods can be added to the discussion:
TM-NET: Deep Generative Networks for Textured Meshes
Share With Thy Neighbors: Single-View Reconstruction by Cross-Instance Consistency

DeepSVG: A Hierarchical Generative Network for Vector Graphics Animation
DeepCAD: A Deep Generative Network for Computer-Aided Design Models
Computer-Aided Design as Language

**Questions:**

1. Since the MeshGPT is closely related, I think comparisons should be included.
https://github.com/lucidrains/meshgpt-pytorch

2. As mentioned in the weakness part, I am really wondering why end-to-end training can generate better results compared to two-stage method like PolyGen and MeshGPT, especially considering 2D SOTA generation model like the latent diffusion model is two-stage.

3. I am not sure the comparisons between MeshXL and PolyGen/GET3D are fair in Table. 2 since it uses pre-trained MeshXL and fine-tunes it for a specific category. I believe using the same dataset for training is more convincing.

4. Why does the scaling law seem not to apply to the lamp as shown in Table 2?

5. Normals in Fig. 7 seem worse (Columns 1, 3, 4) compared to GET3D.

6. will you release the code and pre-trained models since training MeshXL takes a lot of resources?

**Limitations:**

Limitation is discussed, I do not see any issues.

---

> ### Author Rebuttal · Authors · 2024-08-06
>
> 📝 **W1: Reasons why single stage method works, and analysis on previous works.**
>
> 💡 **A:** We thank the reviewer for helping us improve our paper.
> 1. **The coordinate sequence representation and auto-regressive generation make our method come true**. With a well-defined ordering system, each 3D mesh can be represented by **one unique coordinate sequence**. Additionally, decoder-only transformers have long been proven to excel in auto-regressive sequence generation. Therefore, by establishing a decoder-only transformer trained on the collection of coordinate sequences (3D meshes), our one-stage pipeline is able to generate high-quality 3D meshes.
> 2. **Analysis of existing works**. Though previous methods have achieved initial success in 3D assets generation, they suffer from certain limitations.
>     1. To preserve high-frequency information, the point and voxel representations require dense sampling on object surfaces, which inevitably lead to great redundancy when it comes to flat surfaces.
>     2. The reconstruction-based methods rely heavily on the quality of the generated multi-vew images.
>     3. The VQVAE-based 3D generation methods require a two-stage training strategy, which poses extra challenges in large-scale training.
>
>
> 📝 **W2: Fair comparison with previous methods.**
>
> 💡 **A:** We reproduce MeshGPT with `gpt2-medium` (355m) and re-implement MeshXL (marked as MeshXL$^{ShapeNet}$) by first training the model on all ShapeNet categories before fine-tuning it to a specified category. 📊 Please refer to **the global rebuttal** for exact results. With a similar amount of parameters (350m), our method could achieve better generation results with a higher COV score, a lower MMD score, and a closer 1-NNA score to 50%. After pre-trained on larger datasets, our method can achieve even better generation results.
>
>
> 📝 **W3: Comparison with deformation-based and command-based methods.**
>
> 💡 **A:** The training of MeshXL enjoys a better flexibility comparing to both deformation-based or command-based methods.
> 1. **MeshXL vs. deformation-based methods**. The deformation-based methods build on prior geometry knowledge for great initialization. However, The deformation-based method requires external expert knowledge for good template initialization, which **is difficult to generalize to complex 3D meshes**. However, our MeshXL direct learns from large-scale collection of diverse 3D mesh data.
> 2. **MeshXL vs. command-based methods**. The command-based methods adopt command sequences to represent the creation process of the visual data. However, collecting commands is much harder than collecting the generated results (3D meshes). Additionally, the command space requires careful designs and is often limited (`arc`, `circle`, and `extrude` in DeepCAD) to create only simple objects, while our MeshXL learns directly on large-scale 3D mesh data with great diversity.
>
> 📝 **Q1: Comparison with MeshGPT**
>
> 💡 **A:** See weakness 2.
>
> 📝 **Q2: Reason why end-to-end training leads to better mesh generation results.**
>
> 💡 **A:** The main motivation of our work is to explore an simple and effective mesh representation to support large-scale training. Therefore,
> 1. **We can hardly say whether two-stage methods are better or worse than end-to-end learning**. However, training a good **vector-quantized** tokenizer is challenging [R1]. In latent diffusion, instead of adopting a VQVAE, the denoising diffusion process learns to generate latent codes predicted by a VAE, which learns the feature distribution of the latent codes.
> 2. **Our end-to-end design better supports large-scale training**.
>     1. By representing a 3D mesh with **one unique coordinate sequence**, MeshXL can directly learn from large scale 3D data.
> Meanwhile, MeshGPT requires to train a **vector-quantized** representation of vertex feature. Additionally, PolyGen requires to generate a vertex sequence and then connect the generated vertices into polygons. Therefore, learning a good alignment between different modules is challenging and requires careful supervision.
>     2. Based on the quantitative results on ShapeNet categories in Table 2, MeshXL can produce high-quality 3D meshes with higher COV scores, lower MMD scores, and closer 1-NNA scores to 50% than previous methods.
>
> [R1] Li, Tianhong, et al. "Autoregressive Image Generation without Vector Quantization." arXiv preprint arXiv:2406.11838 (2024).
>
> 📝 **Q3: Fair comparison with existing methods**
>
> 💡 **A:** See weakness 2.
>
> 📝 **Q4: Scaling law does not apply to the `lamp` category.**
>
> 💡 **A:** The limited training data leads to overfitting. After pre-processing the ShapeNet dataset, the lamp subset contains **only 565 samples** for training. Therefore, larger models will easily overfit. Instead, we show in Figure 3 of the main paper that when pre-training on extensive 3D data, MeshXL achieves both lower training and validation loss. Additionally, by evaluating on Objaverse, we also notice that as the model size grows, we achieve better generation results.
>
> | Model | COV$\uparrow$ | MMD$\downarrow$ | 1-NNA | JSD$\downarrow$ | FID$\downarrow$ | KID$\downarrow$ |
> |-|-|-|-|-|-|-|
> |MeshXL - 125m| 39.76 | 5.21 | 67.34 | 26.03 | 17.32 | 4.48 |
> |MeshXL - 350m| 40.79 | 5.20 | 65.68 | 23.71 | 15.14 | 3.33 |
> |MeshXL - 1.3b| **42.86** | **4.16** | **61.56** | **20.99** | **12.49** | **2.94** |
>
>
>
> 📝 **Q5: Normal vector comparison with GET3D**
>
> 💡 **A:** In Figure 7, we show the normals to compare **the smoothness of object surfaces**. The results show that 3D meshes generated by GET3D have rough surfaces with tens of thousands of triangles, while ours depict the 3D shape with much smoother surfaces and less triangles.
>
> 📝 **Q6: Open-sourcing.**
>
> 💡 **A:** We have released our code and pre-trained weights to the community. We will also keep updating for additional features.

---

> > ### Comment · Reviewer_CF59 · 2024-08-13
> > **Reply to the rebuttal**
> >
> > Thanks for your great efforts! After reading the response, some major issues have been addressed well, so I still lean towards positive for the submission. I encourage the author to add these clarifications to the main paper. Thanks!

---

> > > ### Author Response · Authors · 2024-08-13
> > >
> > > We sincerely appreciate your recognition of our work! We will incorporate the valuable feedback from all reviewers to enhance our main paper with additional analysis, comparisons, and evaluations. Thank you once again for your time and effort in reviewing our paper!

---

### Official Review · Reviewer_e1fg · 2024-07-13

**Soundness:** 3
**Presentation:** 3
**Contribution:** 2
**Rating:** 4
**Confidence:** 4

**Summary:**

This paper proposes a way to use LLM to generate polygon meshes. The key idea is to model mesh generation as the next coordinate prediction, using a strategy similar to that of prior work like PolyGen and MeshGPT. The paper has shown the capability to create a mesh with reasonable quality.

I think the key contribution for the paper is that it shows the potential to use LLM style scaling to achieve better mesh generation, while the weakness is that it's unclear how much technical contribution given most techniques are already been explored in prior works like MeshGPT and PolyGen.

**Strengths:**

The paper shows the potential to use large-scale LLM models to produce better mesh generation. Specifically, with more compute and larger models, one can achieve stable training and better mesh generation quality.

**Weaknesses:**

My main concern about the paper is its potential lack of novelty. The key idea of using an Autoregressive model to generate a mesh has been explored in many prior works, such as PolyGen and MeshGPT. However, the writing and the results show that this paper differs technically from existing methods other than running on larger datasets with larger-scale models. At the same time, it's not clear from the writing how this paper handles many technical challenges of modeling mesh as a sequence in a new way, such as permutation invariance of the faces.

Also, I think the paper lacks a proper evaluation of mesh quality. The introduction is set to claim that the generative mesh has better quality and is potentially more suitable for downstream applications compared to modeling other representations, such as point clouds or voxels. However, few metrics are indicating that the generative mesh is of good quality that the artist can edit. For example, what's the ratio of the generative mesh being watertight? How well is the triangulation? Most generative metrics this paper reports concern the shape this mesh represents but not how well this mesh triangulates.

**Questions:**

* Question about the ordering. Is the ordering for creating the sequence unique? how does it related/differ from the ordering polygon?  I think that in L103 the paper should cite PolyGEN for the partition idea.

* L29-35 I believe this positioning risks overclaiming. The auto-regressive model can also have cumulative error issues. It's not entirely clear why auto-regressive mesh generation does not have "great redundancy when representing flat surfaces".

* L105 - what is "polynomial face"? I think L101-106 requires much more detailed descriptions for readers to find it reproduciable.

**Limitations:**

See the weakness section. Other limitations include restricted context length (i.e. mesh has very long context lengths, as many meshes have trillions of triangles).

---

> ### Author Rebuttal · Authors · 2024-08-06
>
> 📝 **W1: The novelty of MeshXL.**
>
> 💡 **A:** The motivation of our MeshXL is to extend the **mesh representations**, **architecture design**, and **training strategy** in existing auto-regressive methods, i.e. PolyGen and MeshGPT, to support **efficient large-scale training on extensive 3D mesh data**. Specifically, our method improves existing methods from the following aspects:
>
> 1. **Training strategy.** Both MeshGPT and PolyGen adopt a two-stage training pipeline. This, however, leads to a complicated training procedure less favors large-scale generative pre-training. However, **MeshXL is a fully end-to-end pipeline, which naturally favors large-scale pre-training**.
> 2. **Mesh representation and architecture**. By modeling 3D mesh data into **one unique coordinate sequence**, our MeshXL only consists of a decoder-only transformer for **next-coordinate prediction**. Meanwhile, PolyGen requires a vertex sequence and a vertex index sequence to turn vertices into polygons. Additionally, MeshGPT adopts a vector-quantized approach by learning a fixed-size codebook to represent the vertex feature before learning to generate token sequences auto-regressively. Though we have introduced a bit more redundancy in our mesh representation, our method enjoys a much simpler architecture design and better supports large-scale training.
> 3. **Data Processing**. The data pre-processing in our MeshXL is also much simpler as we are only required to permute 3D faces into a coordinate sequence. Our simplified data pre-processing increases the total throughput to further support large-scale training. Meanwhile, MeshGPT requires to build a face graph with respect to face connectivity within the 3D meshes to extract face and vertices features.
>
>
> 📝 **W2: Additional metrics for mesh quality assessment.**
>
> 💡 **A:** We will add more metrics for better mesh quality assessment in the revision.
> 1. **How well is the triangulation**. Following suggestions from reviewer XQPQ, we evaluate the aspect ratio, face area, and number of faces for a better evaluation in the following table. Though the meshes generated by our MeshXL have a higher average aspect ratio, we achieve a smaller variance with much less 3D faces. This indicates the **stability** of our generation ability and the **efficiency** of the direct mesh representation. Since we train our MeshXLs only on triangular meshes, long-thin triangles inevitably exist in our training data. In future works, we will co-train our MeshXLs on triangular meshes, 3D quads, and even hybrid representations to reduce the existence of long thin triangles for better generation quality.
> | Method | Aspect Ratio | | | Face Area | | | Number of Faces| |
> |-|-|-|-|-|-|-|-|-|
> | | mean | std. | | mean | std. | | mean | std. |
> | GET3D | 6.27 | 116.03 | | 0.000 | 0.000 | | 27251.80 | 11535.135 |
> | MeshXL - 125m | 10.47 | 16.88 | | 0.031 | 0.096 | | 327.34 | 174.53 |
> | MeshXL - 350m | 10.25 | 16.09 | | 0.032 | 0.099 | | 342.24 | 193.97 |
> | MeshXL - 1.3b | 10.23 | 15.91 | | 0.034 | 0.102 | | 320.36 | 195.43 |
>
> 2. **Watertight meshes**. A watertight mesh does not have any boundary edges. Therefore, it is debatabe whether we should generate watertight meshes in 3D assets generation. Currently, watertightness is mainly required to perform physical simulation or turn 3D meshes into implicit field and distant functions. However, in 3D assets generation, many common 3D shapes including cloth, terrain, and leaves are not watertight. Furthermore, it is also challenging and inaccurate to specify the interior of a 3D mesh for an reliable evaluation. Therefore, our method mainly focus on establishing a more direct representation for 3D mesh generation.
>
>
> 📝 **Q1: Question about the ordering.**
>
> 💡 **A:** We will cite PolyGen in Line 103, and clarify the relation between our mesh representations and PolyGen's in section 3 .
> 1. We adopt the same ordering system as PolyGen and MeshGPT, which first permutes the vertices within each face cyclically based their coordinates in z-y-x order (from lower to higher), then permutes the faces based on the permuted coordinates from lower to higher. Since there are no identical polygons in a 3D mesh, the ordering strategy will create **one unique sequence** for each 3D mesh.
> 2. **Relation to the ordering in PolyGen**. We adopt the same vertex representation as PolyGen, but a **different mesh representation**. Our MeshXL and PolyGen both represent vertices with discrete coordinates. However, **MeshXL represents a 3D mesh only with an ordered coordinate sequence**, while PolyGen decouples the vertices and polygons in 3D meshes, and represents each 3D mesh with two sequences, i.e., the vertex sequence and an index sequence to connect the generated vertices. Therefore, our coordinate sequence representation enables us to train an end-to-end model directly on large-scale 3D data.
>
>
>
> 📝 **Q2: Clarifying the claims.**
>
> 💡 **A:** We will clarify our claims and motivations in our revision.
> 1. We will modify the claim into "*comparing to the indirect way of mesh generation, our end-to-end pipeline better supports large-scale pre-training, especially considering the training and data processing efficiency*". We will also do further study to explore better auto-regressive pipelines.
> 2. In our main paper, our intended idea is that the 3D mesh representation has great flexibility that can represent flat surfaces with much less data (i.e. two triangles for a rectangle surface vs lots of points and voxels) and preserve details with more faces in curved surfaces. We will clarify this in our revision.
>
>
> 📝 **Q3: The definition of "polynomial face".**
>
> 💡 **A:** In our paper, a k-sided polynomial face is a polygon with k vertices and k lines. For example, triangles and quadrilaterals are special cases with three and four sides, respectively. We will clarify this in our revision. We have also open-sourced our code to help readers better understand our method.

---

> > ### Comment · Reviewer_e1fg · 2024-08-14
> > **Thanks for the response**
> >
> > Thanks for the detailed response. After reading other reviews together with the rebuttal, I lean to believe that the paper does carry some technical novelties to make mesh generation more scalable - some of which help simplify the pipeline, which is under-appreciated by the community. With that being said, I am willing to change my score toward acceptance.
> >
> > I strongly encourage the authors to revise the paper to bring out the technical contribution needed to simplify the pipeline and make mesh generation more scalable. I would appreciate more detailed discussion on the difference between MeshXL's ordering method and PolyGen's.
> >
> > The evaluation on mesh quality is still unconvincing since the presented metrics can be biased by the fact that the generated meshes are outputted by a model which only sees short meshes. This means that the model can sacrifice other aspects of the mesh qualities, such as being watertight or without self-intersecting faces, just to generate a subset of faces well. I believe this is still the most significant limitation of the current method and would appreciate it if the authors make appropriate acknowledgment of that.

---

> ### Author Response · Authors · 2024-08-14
> **Appreciate the recognition**
>
> We sincerely appreciate your recognition of our work and the valuable feedback to help us improve our paper.
>
> In our revision, we will highlight that our work **paves the way for scaling up training on large-scale 3D mesh data**. Our **mesh representation** turns a 3D mesh into one unique coordinate sequence, which enables us to simplify our **architecture design** into a decoder-only transformer model, facilitating an **end-to-end training pipeline** that does not require sophisticated data pre-processing, careful model design, or complex training strategy and better suits large-scale 3D mesh data.
>
> To further improve our paper, we will include a detailed discussion between MeshXL's mesh representation and that of PolyGen. Additionally, we will also incorporate objective evaluations on the triangulations for mesh quality assessment.
>
> We acknowledge that the potential generation of certain artifacts is a limitation for now. To alleviate the potential occurrence of artifacts, we will keep exploring methods to integrate domain knowledge as filters or reward models. We will also work on co-training MeshXL on triangle meshes, 3D quads, and even hybrid representations to reduce the occurrence of long thin triangles as also mentioned by reviewer XQPQ.
>
> Once again, we thank you very much for your recognition and valuable suggestions from all reviewers to help us improve our work.

---

### Official Review · Reviewer_XQPQ · 2024-07-14

**Soundness:** 4
**Presentation:** 3
**Contribution:** 3
**Rating:** 6
**Confidence:** 4

**Summary:**

This paper addresses the challenge of generating high-fidelity 3D meshes by introducing Neural Coordinate Field (NeurCF), an effective representation for large-scale sequential mesh modeling. The authors present MeshXL, a family of generative pre-trained auto-regressive models, which applies modern large language model techniques to 3D mesh generation. Extensive experiments demonstrate that MeshXL produces high-quality 3D meshes and outperforms existing methods on various tasks. Key contributions include validating NeurCF as a viable representation for 3D meshes, presenting MeshXL as robust base models for conditioned 3D mesh generation, and showcasing MeshXL's superior performance in generating detailed 3D meshes compatible with current texturing methods.

**Strengths:**

1. This paper trains a foundational mesh generation model using extensive datasets from ShapeNet, 3D-FUTURE, Objaverse, and Objaverse-XL, with the addition of data augmentation.
2. It proposes a novel 3D mesh representation that can be encoded as a token sequence, effectively leveraging the capabilities of autoregressive large language model approaches.
3. The paper establishes a fair evaluation metric, considering both the generation score (as shown in Table 2) and the 3D mesh quality from a graphics perspective (as shown in Table 3).

**Weaknesses:**

1. This method does not appear to incorporate domain knowledge from traditional remeshing techniques to ensure correct connectivity between different components, avoid self-intersections, and prevent flipping.
2. In the user study, more objective metrics for measuring mesh quality should be considered. For instance, in downstream tasks like ray tracing, long thin triangles should be avoided, and aspect ratio can be used to measure how thin these triangles are.

**Questions:**

1. How does this method address common mesh surface artifacts in modeling, such as ensuring correct connectivity between different components, avoiding self-intersections, and preventing flipping?
2. In Section 4, we generate triangles within “<tri> · · · </tri>” and quadrilaterals within “<quad> · · · </quad>”. However, what is the form of the output in the results presented in the paper? Should these sequences of triangles and quadrilaterals be generated separately or can they be combined in the final meshing result?

**Limitations:**

Please check weakness

---

> ### Author Rebuttal · Authors · 2024-08-06
>
> 📝 **W1: The absence of domain knowledge to prevent potential artifacts**.
>
> 💡 **A:** In our work, we put emphasis on exploring a sequential way to model 3D meshes that better suits large-scale generative training on extensive 3D data. Therefore, the potential generation of surface artifacts is currently the limitation for our method, as well as PolyGen and MeshGPT. However, our MeshXL has the potential to incorporate with traditional methods by treating the domain knowledge as **filters** or even **reward models** to eliminate certain artifacts.
>
> 📝 **W2: Accessing meshes with additional objective metrics**.
>
> 💡 **A:** We will add more objective metrics in our main paper. We calculate the face area and the aspect ratio with respect to the definition: $\text{Aspect Ratio} = \frac{\text{longest edge}}{\text{shortest altitude}}$. Since PolyGen generates polygon meshes rather than triangle meshes, we could not calculate the aspect ratio for PolyGen. From the below table, though GET3D achieves a lower average aspect ratio, it suffers from a higher variance with tens of thousands of faces. Meanwhile, MeshXL achieves a much stable aspect ratio with a larger average face area, indicating that the our MeshXL has the stability to generate high-quality 3D meshes.
>
> | Method | Aspect Ratio | | | Face Area | | | Number of Faces| |
> |-|-|-|-|-|-|-|-|-|
> | | mean | std. | | mean | std. | | mean | std. |
> | GET3D | 6.27 | 116.03 | | 0.000 | 0.000 | | 27251.80 | 11535.135 |
> | MeshXL - 125m | 10.47 | 16.88 | | 0.031 | 0.096 | | 327.34 | 174.53 |
> | MeshXL - 350m | 10.25 | 16.09 | | 0.032 | 0.099 | | 342.24 | 193.97 |
> | MeshXL - 1.3b | 10.23 | 15.91 | | 0.034 | 0.102 | | 320.36 | 195.43 |
>
> **How to alleviate long thin triangles**. Currently, we train our MeshXL on 3D triangular mesh data. Therefore, long thin triangles inevitably exist in our ground truth training data, for example, the legs of tables or chairs. To alleviate the generation of long thin triangles, we will train our method on 3D quads in the future, which has been proved to be an efficient and common mesh representation in the 3D design industry, and could avoid the generation of long thin triangles.
>
>
> 📝 **Q1: Addressing potential artifacts in the generated meshes.**
>
> 💡 **A:** See Weakness 1.
>
>
>
> 📝 **Q2: The sequential mesh representations.**
>
> 💡 **A:** We will clarify the sequential mesh representations in the revision.
>
> 1. **We train only on triangular meshes in our paper**. In order to accelerate the data processing and training procedure, we choose an easier setting by turning all 3D meshes into triangular meshes to validate that the **next-coordinate generation** is capable of high-quality 3D mesh generation. Thus, the output of our method is always represented with $\text{<tri>} (x,y,z),(x,y,z),(x,y,z); ... \text{</tri>}$.
> 2. **We can easily extend MeshXL to hybrid representations**. In our paper, we present “<tri> · · · </tri>” and “<quad> · · · </quad>” as a potential that our method can be extended to both triangular meshes, 3D quads, and even the hybrid representation. We will further extend our work on the hybrid representation to alleviate the existence of long thin triangles as mentioned in weakness 2.
> 3. If there are both triangles and quadrilaterals in a 3D mesh, our method also follows the ordering strategy introduced in line 111 - 114, which first permutes the vertices within each face cyclically based their coordinates (z-y-x order, from lower to higher), and then order the faces based on the permuted coordinates from lower to higher. Therefore, **a 3D mesh with both triangles and quadrilaterals can also be represented by one unique sequence**. Thus, even with the hybrid representations, our MeshXL can generate these 3D meshes in one sequence.

---

### Author Rebuttal · Authors · 2024-08-06

We thank all reviewers for approval: 1. a **novel and elegant 3D mesh representation** (R1, R4) that 2) **effectively leverages LLM approaches** (R1, R2) for **end-to-end large-scale** training (R1, R3), and 3) a **stable training and better mesh generation quality by scaling up** (R2, R3) supported by 4) a **fair and convincing evaluation** (R1, R4) covering both generation score and mesh quality from a graphics perspective (R1). (R1 - Reviewer XQPQ, R2 - Reviewer e1fg, R3 - Reviewer CF59, R4 - Reviewer AHoX)

We also thank all the reviewers for their valuable suggestions to help us improve our paper. We will address your concerns and revise the paper carefully. We have provided **additional visualization results in the attached pdf file**. Please find our item-to-item responses to your concerns below.

**Motivation and Novelty**. To better support large-scale training, we verify that we can represent a 3D mesh into **one unique coordinate sequence** based on a well-defined ordering strategy. With this simple mesh representation, our MeshXL only requires a single decoder-only transformer for sequence modeling. Comparing to prior two-stage works, **our MeshXL is an end-to-end single-stage pipeline** does not require sophisticated data pre-processing, careful model design, or complex training strategy. Therefore, our MeshXL better suits learning from large-scale 3D data.

**Additional baseline and evaluations**. In the following table, we re-produce the MeshGPT with `gpt2-medium` (355m) using the third-party implementation. We also follow the setting from previous works by pre-training MeshXL (350m) on ShapeNet before fine-tuning to specified categories, which is marked as (MeshXL$^{ShapeNet}$) in the following table. One can see that our method consistently achieves better results than all previous methods. Comparing to MeshGPT with a similar amount of parameters (~350m), our method achieves a higher COV score, a lower MMD score, and a closer 1-NNA score to 50%.

| Category | Method | COV$\uparrow$ | MMD$\downarrow$ | 1-NNA | JSD$\downarrow$ | FID$\downarrow$ | KID$\downarrow$ |
|-|-|-|-|-|-|-|-|
| **Chair** | PolyGen | 7.79 | 16.00 | 99.16 | 228.80 | 63.49 | 43.73 |
| | GET3D | 11.70 | 15.92 | 99.75 | 155.25 | 67.84 | 42.10 |
| | MeshGPT | 42.00 | 4.75 | 69.50 | 55.16 | 39.52 | 8.97 |
| | MeshXL - 125m | 50.80 | **3.11** | 56.55 | 9.69 | 28.15 | 1.48 |
| | MeshXL$^{ShapeNet}$ - 350m | 47.94 | 3.26 | 57.54 | 13.42 | 29.14 | 1.79 |
| | MeshXL - 350m | 50.80 | 3.17 | **55.80** | 9.66 | 28.29 | **1.39** |
| | MeshXL - 1.3b   | **51.60** | 3.23 | **55.80** | **9.48** |  **9.12** | 1.84 |
| **Table** | PolyGen | 44.00 | 3.36 | 67.20 | 25.06 | 54.08 | 14.96 |
| | GET3D | 16.80 | 10.39 | 91.90 | 226.97 | 67.65 | 34.62 |
| | MeshGPT | 34.30 | 6.51 | 75.05 | 92.88 | 53.75 | 7.75 |
| | MeshXL - 125m | 51.21 | 2.96 | 57.96 | **12.82** | 42.55 | **0.92** |
| | MeshXL$^{ShapeNet}$ - 350m | 49.75 | **2.90** | **54.72** | 13.75 | 44.92 | 1.80 |
| | MeshXL - 350m | 49.70 | 3.07 | 56.10 | 13.64 | 43.43 | 1.27 |
| | MeshXL - 1.3b   | **52.12** | 2.92 | 56.80 | 14.93 | **22.29** | 2.03 |
| **Bench** | PolyGen | 31.15 | 4.01 | 83.23 | 55.25 | 70.53 | 12.10 |
| | MeshGPT | 34.92 | 2.22 | 68.65 | 57.32 | 52.47 | 6.49 |
| | MeshXL - 125m | 54.37 | 1.65 | 43.75 | 16.43 | **35.31** | **0.82** |
| | MeshXL$^{ShapeNet}$ - 350m | 55.75 | **1.46** | **44.64** | **10.66** | 36.81 | 1.48 |
| | MeshXL - 350m | 53.37 | 1.65 | 42.96 | 15.41 | 36.35 | 0.96 |
| | MeshXL - 1.3b   | **56.55** | 1.62 | 39.78 | 15.51 | 35.50 | 1.60 |
| **Lamp**  | PolyGen | 35.04 | 7.87 | 75.49 | 96.57 | 65.15 | 12.78 |
| | MeshGPT | 41.59 | 4.92 | 61.59 | 61.82 | 47.19 | 5.19 |
| | MeshXL - 125m | **55.86** | 5.06 | 48.24 | 43.41 | 34.61 | **0.84** |
| | MeshXL$^{ShapeNet}$ - 350m | 52.74 | **3.39** | 41.15 | **25.03** | 31.18 | 1.06 |
| | MeshXL - 350m | 53.52 | 4.18 | **49.41** | 34.87 | **25.94** | 1.92 |
| | MeshXL - 1.3b   | 51.95 | 4.89 | 47.27 | 41.89 | 31.66 | 0.99 |

---

### Author Response · Authors · 2024-08-13

Dear Reviewers,

Thank you for your valuable feedback on our paper "MeshXL: Neural Coordinate Field for Generative 3D Foundation Models". We hope our responses have addressed your concerns to your satisfactory. If you have any further concerns, please let us know during the discussion session.

Thank you again for your valuable time and effort!

Best regards,

All authors

---

### Author Response · Authors · 2024-08-14

Dear Reviewers,

We sincerely appreciate your constructive feedback on our paper, "MeshXL: Neural Coordinate Field for Generative 3D Foundation Models." As the discussion session is about to end, we apologize if we did not fully address your concerns in our discussions. If there are additional concerns or anything you would like us to address, please let us know.

We deeply appreciate your valuable time and effort.

Best regards,

The Authors

---

### Decision · Program_Chairs · 2024-09-25

**Decision:**

Accept (poster)

**Comment:**

The paper received three positive reviews before rebuttal. After rebuttal, all reviewers became positive about the paper. Therefore, the decision is to accept the paper. The paper introduces an explicit coordinate representation with implicit neural embeddings, enabling large-scale sequential mesh modelling. The approach is justified through extensive experiments and its usefulness is also verified through various down-stream applications. In the final version, it is recommended that to take the reviewers comments into consideration and revise the paper based on promise in the proposal. In particular, please articulate the difference between MeshXL's ordering method and PolyGen's.

Although this is not required to be discussed in the paper. One question for the mesh representation is that a single geometric shape can have many mesh discretizations. How would the proposed approach generate the same geometry but different meshes. We do not have this issue in language models or part-based representations of 3D shapes.